# Waypoint Transfer Module between Autonomous Driving Maps Based on LiDAR Directional Sub-Images

**DOI:** 10.3390/s24030875

**Published:** 2024-01-29

**Authors:** Mohammad Aldibaja, Ryo Yanase, Naoki Suganuma

**Affiliations:** The Advanced Mobility Research Institute, Kanazawa University, Kanazawa 920-1192, Japan; ryanase@staff.kanazawa-u.ac.jp (R.Y.); suganuma@se.kanazawa-u.ac.jp (N.S.)

**Keywords:** lane graph, waypoints, mapping system, autonomous vehicles, LiDAR, autonomous driving, slam

## Abstract

Lane graphs are very important for describing road semantics and enabling safe autonomous maneuvers using the localization and path-planning modules. These graphs are considered long-life details because of the rare changes occurring in road structures. On the other hand, the global position of the corresponding topological maps might be changed due to the necessity of updating or extending the maps using different positioning systems such as GNSS/INS-RTK (GIR), Dead-Reckoning (DR), or SLAM technologies. Therefore, the lane graphs should be transferred between maps accurately to describe the same semantics of lanes and landmarks. This paper proposes a unique transfer framework in the image domain based on the LiDAR intensity road surfaces, considering the challenging requirements of its implementation in critical road structures. The road surfaces in a target map are decomposed into directional sub-images with X, Y, and Yaw IDs in the global coordinate system. The XY IDs are used to detect the common areas with a reference map, whereas the Yaw IDs are utilized to reconstruct the vehicle trajectory in the reference map and determine the associated lane graphs. The directional sub-images are then matched to the reference sub-images, and the graphs are safely transferred accordingly. The experimental results have verified the robustness and reliability of the proposed framework to transfer lane graphs safely and accurately between maps, regardless of the complexity of road structures, driving scenarios, map generation methods, and map global accuracies.

## 1. Introduction

Accurate map generation enables safe autonomous driving because different autopilot modules, such as localization, perception, and path-planning, use maps [1,2]. Maps include a topological illustration of environments as well as semantic information (lane graphs) about road structures, as shown in Figure 1. Hence, autonomous driving can be safely conducted by locating vehicles in the real world with respect to the road structures using topological maps and enabling autonomous maneuvering in relation to semantic information using corresponding vector maps.

The general approach to building topological maps is to precisely estimate the vehicle trajectories and then accumulate the sensory data accordingly [3]. The trajectory estimation can be achieved using GNSS/INS-RTK (GIR) systems, SLAM technologies, or Dead Reckoning (DR) based on the vehicle velocity and time interval between positions [4,5,6,7,8]. The topological illustration can be achieved by encoding dense sensory representations, such as 3D point clouds and 2D grid maps, or sparse descriptions, such as feature-based maps [9,10,11,12,13]. The road semantics are included in vector maps to provide lane width, road width, number of lanes, driving directions, traffic signal positions, lane relevancies, speed limits, and so on [14]. This information is usually encoded in a set of waypoints along the driving areas inside road segments and used by the path-planning module during autonomous driving [15,16]. The waypoints are assigned to the topological maps by either manually conducting a labeling process using nicked eyes or automatically extracting them using lane lines and landmark detection techniques [17]. LiDAR road surface images are used in [18] to train a deep learning network to extract the geometry and topology of the underlying lane network on highways. In [19], the waypoints (mentioned anchor points in lane graphs) are estimated by understanding the scene in urban driving using deep learning. A LaneGraph Net is designed and trained using multimodal bird’s-eye view data, including LiDAR road surface images, to generate directional lane graphs. A more sophisticated approach is proposed in [20] to train an autoencoder-based network using the motion patterns of traffic participants to predict the lane graphs in aerial RGB images.

Despite the above impressive achievements and the fact that road semantic information is usually fixed and less subject to change, it can be inferred that waypoint generation is a highly costly process, and a module to transfer the lane graphs between maps should be proposed to avoid regenerating the waypoints. This is because the global accuracy of the topological maps may differ after collecting new data to extend/update the encoded areas to contain new road segments or rebuilding by SLAM technologies to improve the map accuracy. Therefore, the previously used vector maps should always be readjusted to match the position differences in the newly generated topological maps. Accordingly, this paper proposes a framework to automatically transfer waypoints between maps using road surface images. Furthermore, the requirements to deal with challenging environments and road structures that may produce misalignments between the scanned road segments from different directions are analyzed and considered in the implementation strategy. The experimental results have verified the reliability and robustness of the proposed framework to readjust the waypoints regardless of the reasons for changing the global accuracy and the complexity of the road structures. This has been proven using map data collected inside tunnels in opposing directions on wide roads with multiple lanes and underpasses with different driving directions, as highlighted in the result section.

## 2. Previous Work

We have proposed a unique mapping framework since 2016 to generate LiDAR maps in the image domain based on road reflectivity instead of the conventional methods in the 3D point cloud domain [3]. This is because the road surface is less subject to changes compared to the surrounding 3D environmental features. Therefore, we designed a node strategy to convert the vehicle trajectory into images in the Absolute Coordinate System (ACS). A 3D point cloud is converted into a 2D intensity frame, and the frames are accumulated in an image with respect to a reference position in ACS to represent the reflectivity of the road surface and the surrounding environment at a height of 30cm. The accumulation process is terminated when the number of pixels in the intensity image exceeds a threshold. Accordingly, a node is added to the map, and the accumulation process is then reinitialized to encode another road segment and extend the map with more nodes, as shown in Figure 2a. Each node is then divided into a set of sub-images with fixed height and width to justify the representation in the image domain. The sub-images are given XY-IDs based on the coordinates of the reference position and the pixel resolution, as indicated in Figure 2b. Thus, these sub-images represent the topological map, and the IDs are used to retrieve the roads around the vehicle during autonomous driving.

A robust Graph SLAM (GS) module has been then implemented to improve the map accuracy in challenging environments such as long tunnels, dense trees, high buildings, and underpasses, as illustrated in Figure 3a–c [7,21]. The essence of the idea is to convert the vehicle trajectories into road surface images, explore the relationships between these images in the real world, and then relocate the images in ACS to recover the consistency and coherency of the road representation. Dealing with nodes in the image domain has facilitated the steps to implement GS and achieve impressive results compared with the conventional methods in the 3D point cloud domain [22,23]. For example, constituting relationships between images instead of vehicle trajectories, detecting revisited areas between images instead of vehicle positions, compensating accurately the relative position errors in the XY plane using dense image matching techniques instead of applying iterative registration techniques on sparse 3D point clouds, as well as facilitating the design of the SLAM cost function to optimize the road consistency in the XY plane instead of optimizing the xyz vehicle trajectories in the 3D point cloud domain. Accordingly, the proposed GS has robustly increased the scalability of the mapping module to generate precise maps in challenging environments, as demonstrated in Figure 3c.

The above stages have significantly addressed the process of generating topological maps for autonomous vehicles with clear and accurate representations of the road surface. This enables us to precisely create and label waypoints over the images and assign the road semantic information with respect to the landmarks. On the other hand, the functionality to transfer the previously labeled vector maps after generating new topological maps is very necessary and important to automate the entire map-building process. Thus, the next section introduces the fundamental assumptions of the implementation process.

## 3. Implementation Strategy

The waypoints (lane graph) of a reference map can be transferred to a target map using two strategies. The first strategy is to keep the vector map fixed and bring the global position of the target topological map to the reference map in ACS. The second tactic is to modify the global position of the waypoints according to the target map. The first strategy is more complicated because of the requirement to implement a technique to remap the target topological map to the reference map. This prevents updating the global position accuracy of the road surface by collecting new data and imposes to a shift of the new road segments to the reference map in order to preserve road consistency. Therefore, the transfer module is implemented in accordance with the second strategy.

The proposed mapping system in the image domain is sufficient to implement a transfer module for lane graphs. This is because of encoding road surfaces and illustrating the printed landmarks in the intensity sub-images. Thus, the semantic information can be transferred by calculating the *xy* translations between road surface images in the common areas. Accordingly, image matching techniques can be applied to estimate the translations and the relevant covariance errors of the matching result. However, a few assumptions should first be considered to optimize the implementation process and the relevant requirements, as follows:The reference topological map is dense in representing the road surface, as shown in Figure 4a. This is because the corresponding vector map is expected have been previously created over the topological map.The reference and target maps are accurate in representing the road context without local ghosting or distortions in landmarks, as illustrated in Figure 4a,b. This means that the quality of the two maps has been checked and confirmed in advance.Relative global position errors may occur between maps in ACS at many road segments with different values. This is because different methods are used to generate maps, such as GIR, GS, or DR, as well as the effects of environmental and road structures, as implied in Figure 4c.The common areas between the reference and target maps should initially be estimated to apply the matching technique to the corresponding images.The non-shared waypoints in the reference map should also be transferred to the target map. In other words, the target map may not encode all topological areas in the reference map, and these missing areas might be added later.

Based on the above assumptions, the sub-image level is the optimal and simplest implementation domain to transfer the waypoints. This is because sub-images are the final product in the mapping process and represent the road surface densely and precisely regardless the generation methods. Accordingly, the next section illustrates the implementation steps and the relevant technical issues.

## 4. Technical Issues in Challenging Environments

The common areas between the reference and target maps can be detected based on the shared XY-IDs of sub-images. The phase correlation technique is then applied between the shared sub-images to estimate the *xy*-translations of the road surfaces in ACS and transfer the existing waypoints to the target map, as illustrated in Figure 5 [24,25,26].

The above basic approach provides significant results in open-sky environments and especially in the sub-images that encode road surfaces without partial or full separations of road shoulders. The separation may occur due to the complexity of the road structure, such as high middle barriers, bridge columns in underpasses, and independent tunnels. This may create misalignments in the longitudinal direction as well as gaps in the lateral direction between road shoulders, as illustrated in Figure 6a. These two events do not affect the localization accuracy because it is important to estimate the relative position errors of the vehicle inside the map locally. On the other hand, they can vary according to generation methods such as GIR, DR, and GS. This leads to different representations of the shared road structures between the reference and target maps. Accordingly, the matching results will be true and safe to transfer the waypoints in a road shoulder and false in the other shoulder with wrongly placing the relevant waypoints, as shown in Figure 6b,d. These wrong semantics of landmarks may lead to wrong trajectories generated by the path planning module and drift the vehicle in the real world, especially at junctions or on narrow roads. Thus, the road shoulders should automatically be separated and independently encoded in the sub-images to enable safe transfer of the waypoints in ACS.

## 5. Directional Sub-Image Based Waypoint Transfer Module

Generating sub-images based on the road orientation is the optimal solution to safely transfer waypoints to the target map and guarantee the same semantic representation of the landmarks compared with the reference topological map. According to the illustrated issue in Figure 6d, the lanes in the same driving directions should be combined in the same sub-image, whereas a different sub-image should automatically be decomposed to encode the opposite lanes (road shoulder). Thus, Yaw-IDs based on the driving direction are given to identify the sub-image in ACS.

A vehicle must move in a lane direction with a smooth change in the heading angle during the mapping data collection phase. However, the heading angle might be changed at turns and intersections considerably. In lane changes and overtaking maneuvers, the typical change in the Yaw angle is less than 70° due to the constraints in the road structure, the traffic flow, and the vehicle dynamics. In contrast, the heading angle changes to nearly 90° at intersections and 180° to conduct U-turns. Therefore, each sub-image is assigned with two YawIDs to represent the vehicle angles of entering $\varphi_{entry}$ and leaving $\varphi_{exit}$ of the encoded road surface with respect to a threshold α = 70° as in Equation (1):(1)$$\varphi_{entry}=\theta_{entry}/\propto ,{\;\varphi }_{exit}=\theta_{exit}/\propto$$
The YawIDs are used to increase the density of the road surface representation by merging the sub-images in the same direction at loop closures in the target map, as well as to automatically and independently separate encode the other directional road surfaces at the same area, as illustrated in Figure 7, for two opposing lanes in the first row and two maps with slightly different driving scenarios in the second row. Based on the shared XYIDs between the reference and target maps, a full sub-image in the reference map is matched to the corresponding directional sub-images in the target map using Phase Correlation to estimate the *xy* translation offsets in each direction. Accordingly, the waypoints in the reference map should be transferred to the target map based on the YawIDs of the matched directional sub-images, as detailed in the next paragraph.

Suppose that a set of waypoints in a sub-image is given with the corresponding links to represent the connections and the lane relevancy, as illustrated in Figure 8. Each waypoint *X* is identified in ACS by three coordinates *x*, *y*, and φ. The waypoint candidates to represent the entering and leaving vehicle positions inside the driving area can be initially determined near the image edges’ coordinates using the two YawIDs. The links are then used to check the possibility of sequentially connecting a pair of candidates through the internal waypoints and create the corresponding regular vehicle trajectory *Tr*. Suppose that two candidates XTrentry and XTrexit are considered to check the connectivity. A set of *i* waypoints that are reachable from XTrentry can be obtained (if a link exists) and described by Equation (2).
(2)$$X^{i}=\left(\begin{array}{c} x_{i}\\ y_{i}\\ \varphi_{i} \end{array}\right)\;(i=1,2,3\cdots )$$
The relative angle between XTrentry and XTri waypoints is then calculated by Equation (3).
(3)φentryi=tan−1⁡xisin⁡φentry−φi+yicos⁡φentry−φi+yentryxicos⁡φentry−φi−yisin⁡φentry−φi+xentry
The *i*th waypoint XTrT which satisfies the minimum difference with the relative yaw angle φentryexit between XTrentry and XTrexit is then selected and added to the trajectory using Equation (4).
(4)XTrT=argminφentryexit−φentryi
The trajectory is then expanded by repeating the above steps with respect to the last added waypoint XTrT until reaching XTrexit. Accordingly, the virtual vehicle trajectory is constructed and the relevant waypoints are specified, as demonstrated in Figure 8b. The created trajectory is supposed to resemble the real/exact trajectory that was used to collect the mapping data on the target map. On the other hand, the virtual trajectory may slightly differ by conducting lane changes or using different parallel lanes in the target map at wide/multiple-lane intersections. These differences do not affect the transferring process because they share the same properties of waypoints in terms of orientation and context, as illustrated by the three green trajectories in Figure 8b.

At each matching process, the trajectory between the two YawIDs of a target directional sub-image is created in the reference sub-image, and the corresponding waypoints are transferred to the target map using the estimated *xy* offsets. The other waypoints that possess the same orientations, such as parallel lanes, are also transferred, as they can be observed along the green trajectories in Figure 8b. After matching all directional sub-images in the target map, the remaining waypoints in the reference map are transferred based on the offsets of the nearest transferred waypoints.

## 6. Results and Discussion

### 6.1. Setup and Experimental Platform

Two vehicles have been used to collect the data in Tokyo and Kanazawa cities in Japan. The vehicles have the same sensor configuration, including the LiDAR Velodyne 64 and GNSS/INS-RTK systems of Apploniax PosLv 220 [27,28], as shown in Figure 9. The sensory data flow via an internal LAN network and are stored in a processing unit in the trunk. The vehicle is manually driven during the data collection phase, and the trajectory is then post-processed by Apploniax software (LV 220) to accurately estimate the vehicle positions. Accordingly, the map is created by accumulating the LiDAR point clouds to represent the road surface, as in [3]. The waypoints are then manually labeled over the road surface images to encode information such as the global position, driving direction, regular velocity, width of lanes, and type of surrounding landmarks. The waypoints are gathered in groups to represent the road context and the lane relevancy. This process is done by creating a set of links that connect the waypoints of a local road segment and describe the regulations with the neighboring groups. Thus, the possibility of conducting autonomous maneuvers such as lane changes, overtaking, U-turns, and turns at junctions is determined by the relationships between the links and the relevant details of the waypoints.

In order to emphasize the reliability and robustness of the proposed system, two challenging courses have been selected in Kanazawa and Tokyo. The courses contain tunnels, wide roads, multiple lanes in each direction, traffic junctions with multiple terminals, and middle road barriers. Furthermore, the courses have been scanned using different vehicles, driving scenarios, and map generation methods (GIR and GS). Moreover, each course has a reference map that was generated in 2021 and used to conduct autonomous driving on many trails without any technical problems regarding the localization accuracy and the path-planning maneuver with respect to the road structure and the traffic flow. The necessity to update the maps has emerged in 2023 due to the considerable changes in landmarks or road texture, such as integrating a thermal layer. Due to the complex road structures, considerable changes in the global accuracy between the reference and target maps have been observed at different road segments. However, the proposed system has robustly and safely transferred the waypoints as provided in the next sections.

### 6.2. Waypoint Transferring between Two Generated Maps by GNSS/INS-RTK System

The first course is in Kanazawa and is illustrated in Figure 10a by 336 nodes that are represented by the top-left corners in ACS. The course encodes the map between Kanazawa University and the seaport through three short, independent tunnels in opposing directions. Moreover, most of the road segments consist of multiple lanes in each driving direction with middle vegetation separators. This limits the LiDAR’s capabilities to sense the opposite lanes and leads to many intersections joining the road shoulders. The reference map was generated in 2021 using the GIR system and used frequently to conduct autonomous driving, as shown in Figure 10b. We observed that the opposing tunnels have global longitudinal misalignments in ACS due to the obstruction of the satellite signals. However, these misalignments did not affect the localization accuracy because of the precise estimation of the vehicle position locally in each tunnel and the recovery of the signal quality at the tunnels’ terminals.

The target map was generated after two years of repainting several road segments with new landmark patterns, as illustrated in Figure 10c. We observed that the target map has better global accuracy and localizes the opposing tunnels in better alignment. In addition, we figured out that the reference map contains lateral deviations between road shoulders near the seaport, as illustrated in the second row in Figure 10d,e. However, these events may also occur in the target map because we scanned the course again and a lateral deviation has been observed at the same segment as shown in the third row in Figure 10d,e. The enlarged patches in Figure 10e emphasize that the deviation is not fixed and differs based on the GIR’s signal quality and the traffic flow.

Figure 11 shows two cases of encoding a signal road shoulder inside the first and second tunnels. Figure 11a shows the combined map image by the GIR system and the longitudinal deviation between the reference (red) and target (green) maps. The two sub-images have the same YawIDs, and Phase Correlation provides accurate offsets to safely transfer the contained waypoints to represent the semantics of the same positions and landmarks, as demonstrated in Figure 11b–d.

Figure 12a illustrates the two cases mentioned in Figure 10d to represent the two road shoulders inside the first tunnel and near the seaport, respectively. Figure 12b shows the results of applying Phase Correlation between reference and full target sub-images to estimate the translation offsets. Accordingly, a road shoulder is perfectly aligned, whereas the other shoulder is misaligned due to the different global position errors. Therefore, the transferred waypoints in the misaligned shoulder considerably represent wrong semantic information compared to the reference map, as demonstrated in Figure 12c,d. The proposed system has separated the road shoulders based on the YawIDs and generated two directional sub-images, as demonstrated in Figure 13. The two directional sub-images created a unique matching pattern with the reference sub-image. Consequently, Phase Correlation provided accurate offsets to safely transfer the relevant waypoints to the target map planes, as illustrated in Figure 13b–d.

### 6.3. Waypoint Transferring between Two Generated Maps by Graph SLAM

The second course encodes the waterfront area of Tokyo (Odiba). Odaiba is a challenging environment for generating maps because of the existence of high buildings and dense trees, as shown in Figure 14a. Moreover, a tram railway with several stations lay all the way above the road. This causes it to considerably deflect and obstruct the satellite signals and leads to unsafe maps generated by the GIR system. We scanned the course to intentionally create many loop closures in different driving directions at many intersections, as indicated by the green links in Figure 14b. Figure 14c shows four samples of the generated map in Odaiba using the GIR system with distortion and deviations in the lateral and longitudinal directions. Therefore, we always use our unique Graph SLAM system [7] to increase the map data accuracy, compensate for relative position errors, and publish the map in ACS while maintaining road constancy and coherency, as illustrated in Figure 14d. Hence, the generated GS maps in Odaiba are locally accurate and may differ in their global positions, as demonstrated in the first two left images in Figure 14e. Moreover, we observed that longitudinal differences between road shoulders may occur between GS maps, as shown in the following two images.

GS has been applied to the reference map in 2020 and then to the target map in 2023 to enhance consistency and coherency in representing the road surfaces. Figure 15a shows a reference sub-image at a traffic junction with the labeled waypoints in the corresponding vector map in Figure 15b. The target sub-image in Figure 15c illustrates some changes in the road pavement and landmarks. The change in the global position between the two sub-images is slightly small, as indicated in Figure 15e, and the waypoints should be transferred accordingly.

The sub-image in the target map has been decomposed into four directional sub-images based on the proposed tactic to provide YawIDs according to the driving scenarios of entering and leaving the junction during the data collection phase. Therefore, the area has been visited six times, as shown in Figure 15e, with indicating the driving trajectories. One can observe that two trajectories exist in the first and second left images. This emphasizes the robustness of the proposed framework to distinguish and combine the mapping data of similar driving scenarios using YawIDs and the threshold α. This increases the density to represent the road surface and robustizes the matching results. Figure 15f illustrates the accurate matching of the road landmarks between the full reference sub-image and the directional target sub-images. This is because encoding a unique road representation in each directional sub-image makes the mismatch extremely rare. Technically, we have not observed any mismatching errors along different courses, and such errors may occur when the full reference and directional target sub-images do not share sufficient features due to a massive difference in the global position or a huge change in representing the road surface. The driving trajectories are reconstructed in the reference sub-image using the YawIDs of the target directional sub-images, and the corresponding waypoints are determined accordingly, as indicated in Figure 15g. This strategy significantly reduces the required data and enables us to not use/save the exact vehicle trajectory. In addition, it allows for the identification of similarly oriented waypoints in the neighboring lanes, as illustrated in the first left image in Figure 15g. Accordingly, the waypoints are precisely transferred from the reference to the target maps, as demonstrated in Figure 15h.

Figure 16 shows traffic junctions with multiple lanes in each direction and different global position errors between reference and target maps, as demonstrated in the enlarged patches. These errors differ locally between lanes at each junction due to the driving scenarios, as previously explained in Figure 10d,e. The proposed system has safely transferred the waypoints from the reference map, regardless of these errors, to describe the same spatial road semantics in the target map. This indicates the robustness and reliability of the proposed system to operate at different road segments and traffic junctions with various driving scenarios.

## 7. Conclusions

A simple and reliable framework has been proposed to transfer lane graphs between reference and target maps safely and accurately. The vehicle trajectory in the target map is converted into road surface images using the node strategy. The unique tactic to decompose nodes into directional sub-images by providing X, Y, and Yaw IDs has enabled us to automatically separate the mapping data based on the driving directions and increase the road density in the same direction. This led to encoding a unique road surface representation in the directional sub-images and obtaining accurate matching with the corresponding full sub-images in the reference map. Moreover, the use of the Yaw IDs to reconstruct a virtual vehicle trajectory in the reference map has significantly enabled the determination of the associated lane graphs in each directional sub-image and increased the accuracy of transferring graphs with the same road semantics in the target map. This significantly led to saving only sub-images without any additional information about the vehicle’s trajectory. The proposed framework has been used in two challenging courses in Kanazawa and Tokyo, including tunnels, multiple lanes, and complex junctions. The results have verified the robustness of safely transferring waypoints using the directional sub-images in the target map, regardless of the different global accuracy that may occur locally in the target map between road shoulders and junction terminals or relatively between reference and target maps. Therefore, the transfer strategy is reliable to operate regardless of the map generation methods, such as GNSS/INS-RTK systems and SLAM technologies, and scalable to be integrated with data from other sensors such as the camera’s road surface images and feature maps.

## Figures and Tables

**Figure 1 sensors-24-00875-f001:**
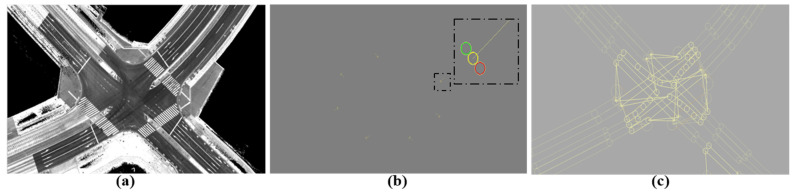
Mapping types for autonomous vehicles. (**a**) Topological Map. (**b**) Vector map: traffic signals with three lights in the horizontal direction. (**c**) Vector map: waypoints.

**Figure 2 sensors-24-00875-f002:**
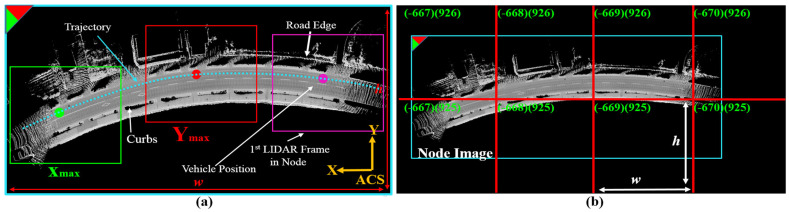
Mapping strategy in node domain. (**a**) Converting vehicle trajectory into a road surface in the image domain and identifying the image as a node in ACS with *xy* top-left corner coordinates. The red and green circles indicate the corner coordinates in maximum XY directions. (**b**) Dividing the node image into a set of sub-images with fixed width and height in the image domain and giving XYIDs.

**Figure 3 sensors-24-00875-f003:**
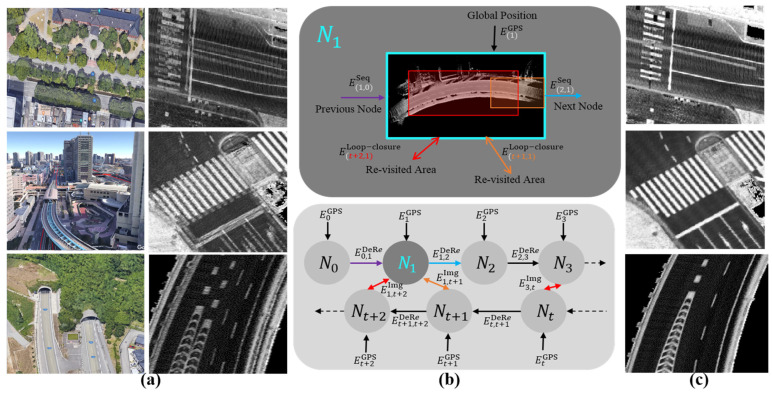
(**a**) Challenging environments for mapping using the Gnss/Ins-RTK system with deviation and distortion in representing the road landmarks. (**b**) A unique Graph SLAM framework in the node domain to improve the map accuracy with showing road surface images as nodes with the relevant connections and loop closures. (**c**) The accurate map generated by Graph SLAM with a clear and precise representation of the road surface.

**Figure 4 sensors-24-00875-f004:**
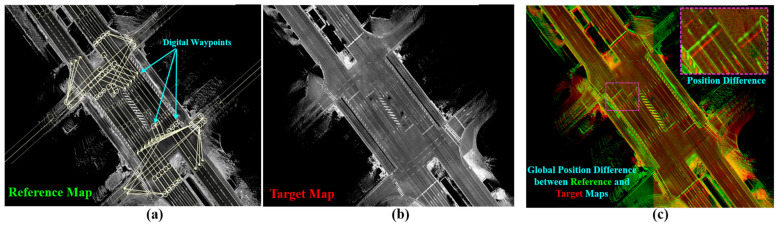
Reference map with manually labeled waypoints representing dense and precise road surfaces. (**b**) Target map representing the same road segment in (**a**) and generated after two years. (**c**) Global position errors in lateral and longitudinal directions between ((**a**): green) and ((**b**): red).

**Figure 5 sensors-24-00875-f005:**
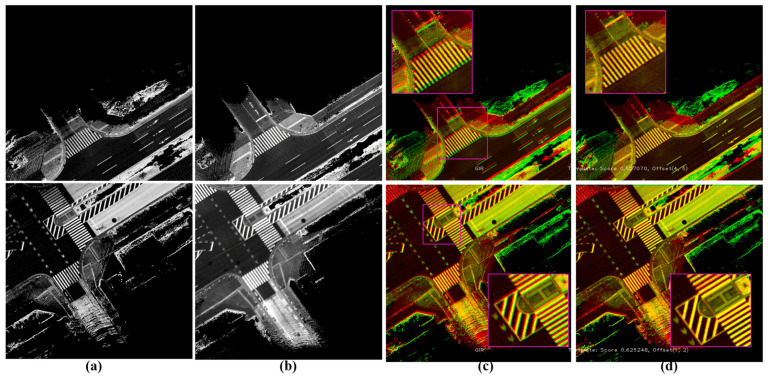
(**a**) A sub-image in a reference map. (**b**) The same road surface encoded in a sub-image in the target map and detected to form a loop closure with the reference map based on the shared XY-IDs. (**c**) Global position errors between reference (red) and target (green) maps represented by distortions in the landmarks in the common areas. (**d**) Matching results by applying Phase Correlation to estimate the *xy* translations while providing accurate combined road representation.

**Figure 6 sensors-24-00875-f006:**
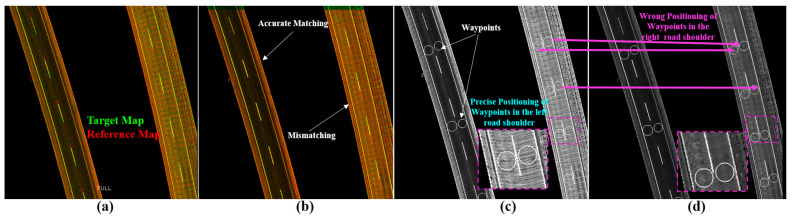
(**a**) Different global errors and misalignments in the longitudinal direction between two independent tunnels in the reference (red) and target (green) maps. (**b**) Matching results by Phase Correlation and a true *xy* offsets in the left tunnel and a wrong estimation in the right tunnel. (**c**) The reference map with the previously labeled waypoints. (**d**) Transferring the waypoints to the target map based on the *xy* translation offsets and wrong placement in the right tunnel to describe the same road semantics in the reference map.

**Figure 7 sensors-24-00875-f007:**
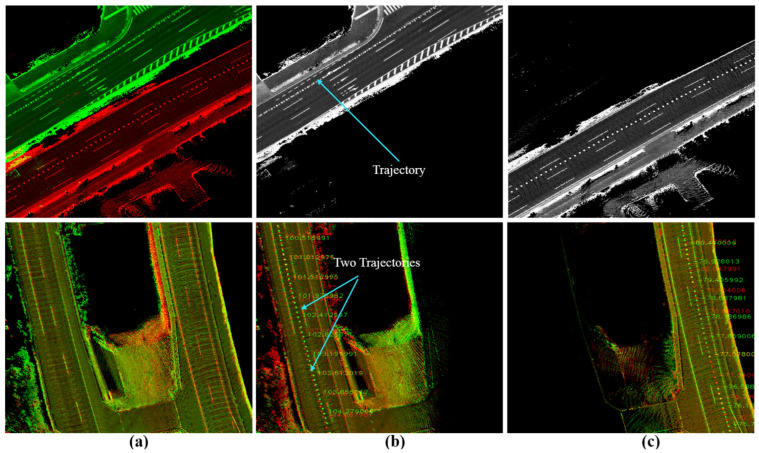
(**a**) Road surface with opposing lanes (the first row scans a road segment in two directions, whereas the second row represents two scans in each direction). (**b**,**c**) Automatic separation of the road shoulders based on Yaw IDs, showing two different trajectories in the second row.

**Figure 8 sensors-24-00875-f008:**
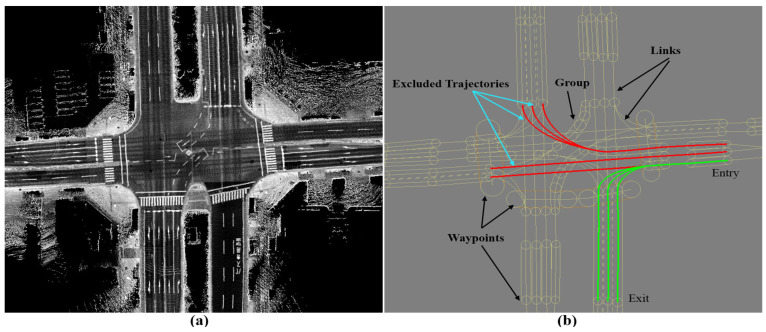
Reconstructing the vehicle trajectory in the reference map based on the Yaw-IDs in the target map. (**a**) Topological road representation by a sub-image in the reference map. (**b**) Waypoints in the reference map and searching on the trajectory to connect the entry and exit waypoints at the image edges that were determined by the Yaw-IDs in the target map. Red trajectories show the false attempts, whereas the greens represent the successful estimation of the vehicle trajectory with the corresponding waypoints.

**Figure 9 sensors-24-00875-f009:**
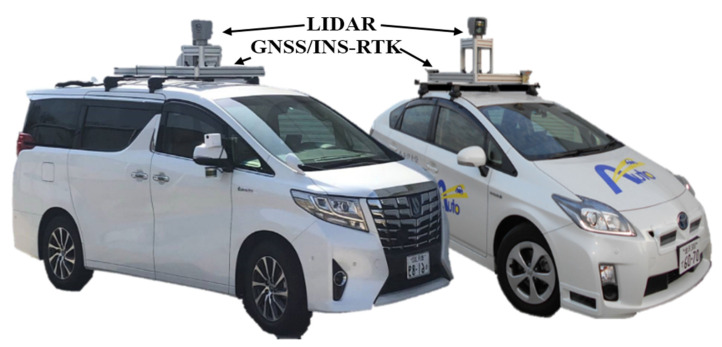
Experimental vehicles with LiDAR and GIR systems.

**Figure 10 sensors-24-00875-f010:**
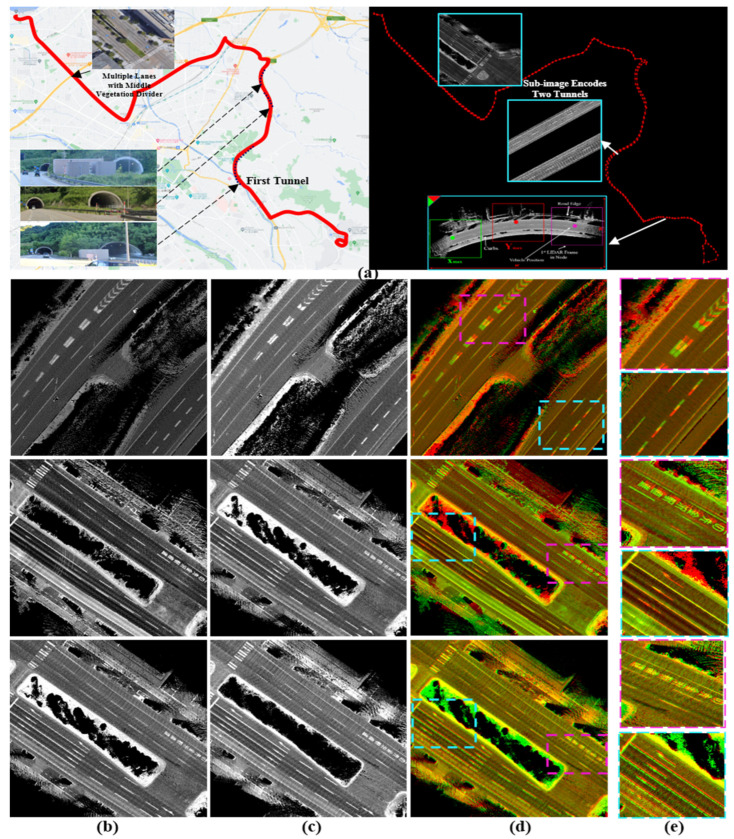
(**a**) Test course between Kanazawa University and the seaport with three sequential tunnels and wide roads with multiple lanes and middle separators. The course is represented by 336 nodes in two directions. (**b**) Reference map: misalignment between road shoulders in the longitudinal direction in the first row and in the lateral direction in the second and third rows. (**c**) Target map. (**d**) Different global errors in the road shoulders between two maps. (**e**) Enlarged road patches in the two road shoulders to indicate the different misalignments (first row: in longitudinal direction in both, second row: in lateral direction in one shoulder (pink), and third row: in longitudinal direction in one shoulder (pink) and in longitudinal direction in the other shoulder (cyan).

**Figure 11 sensors-24-00875-f011:**
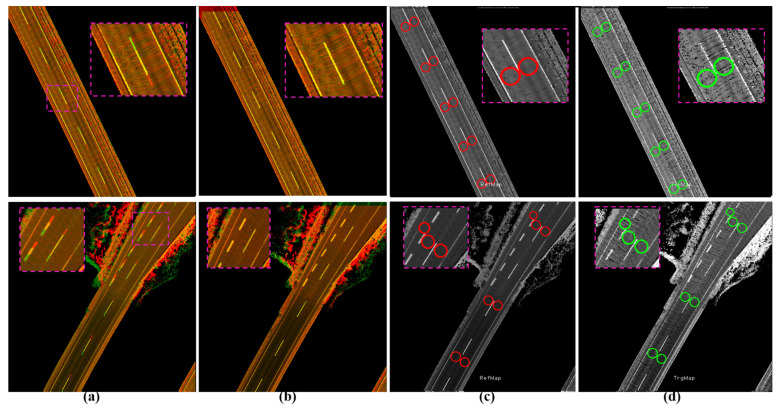
Single-encoded road shoulder (tunnel) in a sub-image based on XY-IDs. (**a**) Global error in the longitudinal direction between two maps generated by the GNSS/INS-RTK system. (**b**) Matching results by Phase Correlation to accurately estimate the offsets. (**c**,**d**) Safe transfer of waypoints from the reference (red circles) to the target (green circles) maps.

**Figure 12 sensors-24-00875-f012:**
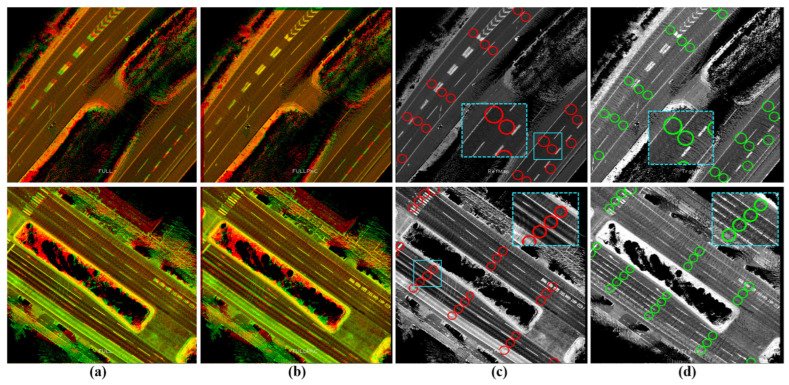
(**a**) Two cases in Figure 10d (second and third rows) showing two sub-images encoding two road shoulders with different global position errors in each direction. (**b**) Compared with (**a**), the Phase Correlation restored the alignment in one direction and increased the misalignment error in the other direction due to the matching of the full road surface images. (**c**) Reference map with the waypoints. (**d**) Transferred waypoints in the target map based on the matching results in (**b**) with the wrong semantic description in the misaligned road shoulders.

**Figure 13 sensors-24-00875-f013:**
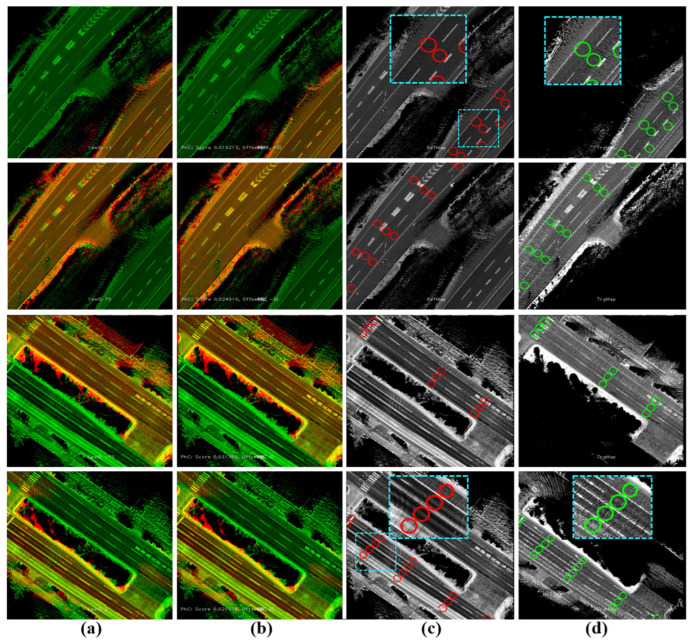
Transferring waypoints using the directional sub-image strategy. (**a**) Full reference map sub-image (green) overlapped by the directional sub-image (red) in the target map with a global position error in the longitudinal direction using the GNSS/INS-RTK system. (**b**) Accurate matching results by Phase Correlation between full reference and directional target sub-images. (**c**) Reference map with waypoints. (**d**) Precisely transferred waypoints in the target image.

**Figure 14 sensors-24-00875-f014:**
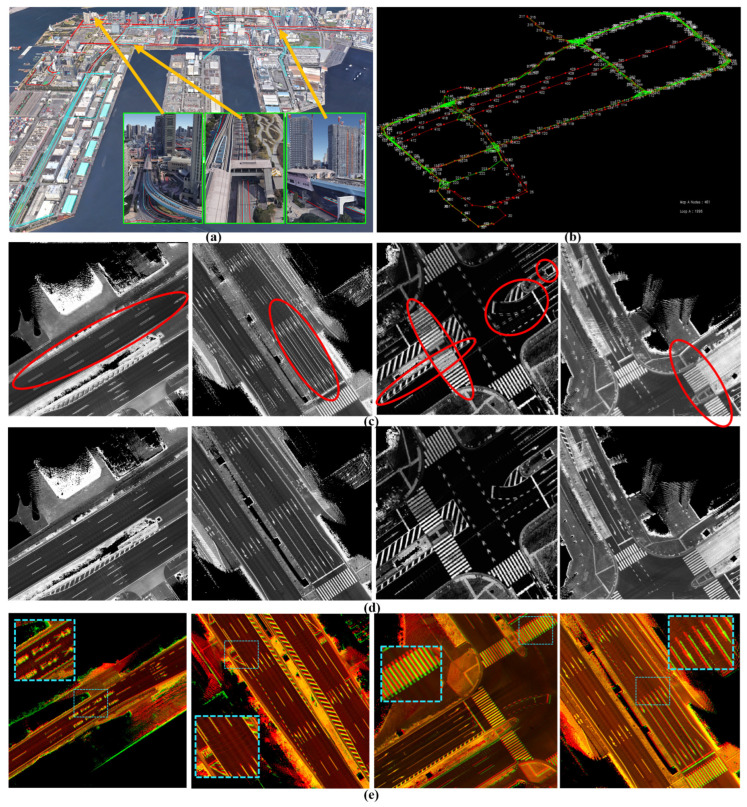
Challenging mapping environments. (**a**) Odaiba area with high buildings, dense trees, and a tram railway with wide roads. (**b**) The course is in the node domain with many loop closures in different driving directions and multiple lanes. (**c**) Generated map by the GNSS/INS-RTK system with distortions and deviations in the road surface due to deflection and obstruction of satellite signals. (**d**) A precise Graph SLAM map. (**e**) Global position errors between two maps generated by Graph SLAM individually.

**Figure 15 sensors-24-00875-f015:**
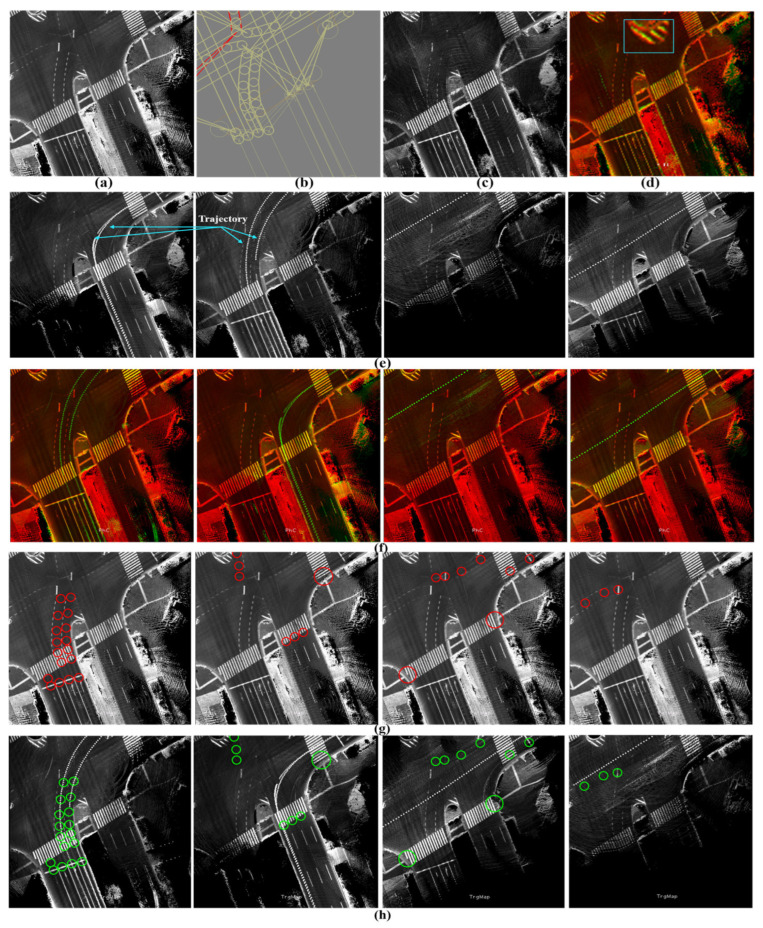
(**a**) Reference map generated by Graph SLAM in 2020. (**b**) The labeled waypoints. (**c**) The target map generated by Graph SLAM in 2023. (**d**) The global position error between the two maps. (**e**) The target map has been decomposed into four directional sub-images based on the Yaw-IDs. (**f**) Accurate matching between the full reference sub-image and the directional target sub-images to estimate the translation offsets in each driving scenario. Yellow landmarks indicate the shared features between the full reference sub-image (red) and the target directional sub-images (green). (**g**) The waypoints were determined by reconstructing the vehicle trajectory in the reference map based on the Yaw-IDs in each directional target sub-image. (**h**) The safe transferred waypoints in the target map.

**Figure 16 sensors-24-00875-f016:**
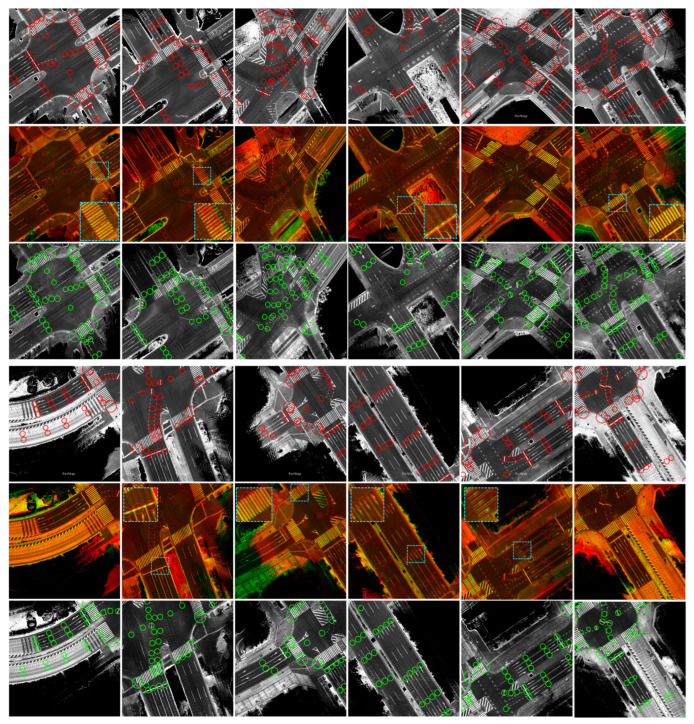
Different illustrations of the accuracy of the proposed system in two groups at different junctions and complex road segments to represent the same road semantics. In each group, the first row shows the reference map (red) with the previously labeled waypoints, the second row shows the combined map images with global errors, and the third row shows the target map (green) with the transferred waypoints. The enlarged patches illustrate an example of the global misalignment between maps.

## Data Availability

The data in this paper belongs to Kanazawa University and not available for public.

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
