# Peer review of "Waypoint Transfer Module between Autonomous Driving Maps Based on LiDAR Directional Sub-Images"

_sensors, 2024, doi:10.3390/s24030875_

Round 1

Reviewer 1 Report

Comments and Suggestions for Authors

The paper proposes a method for transferring waypoints between autonomous driving maps based on LIDAR directional sub-images. The method is easy to understand and implement. The authors conduct experiments to verify the effectiveness of the method.

Despite the broad and important application of the method, the authors do not investigate the related work sufficiently and do not compare the proposed method with existing existing methods in the experimental part, which makes it difficult for me to evaluate the significance of the method.

Comments on the Quality of English Language

The paper is generally easy to read.

Reviewer 2 Report

Comments and Suggestions for Authors

This paper proposed a waypoint transfer module for autonomous driving. It's clear that the focus is on lane graphs transfer between maps.

However, it is difficult to determine the consistency between the framework the paper tried to address and the actual results presented. The paper needs to identify more explicitly whether and/or how much the experimental design as well as the results covered real-world waypoint transfer scenarios. The results seeemed to be more specific compared to the general aim of the paper. Please see my comments below.

First, at the beginning, it seemed like the paper stated explicitly that the results had verified all possible scenarios regardless of complexity (e.g. road structures, driving scenrios,...). However, it's not clear from the results in Section 6 whether those scenarios have been covered.

Second, the paper mostly presented the results in Section 6 pictorially. Even though the figures clearified visually about how the proposed module could achieve, it's not clear how much the proposed module would perform in different road environments. Would it be possible to present the results numerically or statistically? 

Third, the paper needs to define more clearly about the definition of "reliability and robustness" as this seemed to be the main benefit of the proposed module. And also how the results reflect the reliability and robustness.

Also, what are the limitations of the proposed module (if there is)? For example, the proposed module seemed to rely primarily on LIDAR. Was there any limitation in using LIDAR? If there were, what're possible future solutions? 

Could the authors could add comments or discussion from practicallity point of view? For example, if the proposed module were to be implemented in real-world scenarios, what should be taken into account? How could it be integrated with data from other types of sensors, or other autonomous driving systesms in general?   

Comments on the Quality of English Language

There are two proposed types of language adjustment.

1) There overall explanation particularly the transition between sections and more discussion on the results. 
2) There were some small errors in the paper. It should be revised more carefully before resubmission. The errors that could be spotted were found on rows 97, 101, 112, caption of Figure 12. Sentences after a (.) should begin with capitcal letters. 

Reviewer 3 Report

Comments and Suggestions for Authors

In this paper, a unique transfer framework in the image domain based on the LIDAR intensity road surfaces is proposed. The road surfaces in a target map are decomposed into directional sub-images with X, Y and Yaw IDs in the global coordinate system. The XYIDs are used to detect the common areas with a reference map whereas the Yaw IDs are utilized to reconstruct the vehicle trajectory in the reference map and determine the associated lane graph. The directional sub-images are then matched to the reference sub-images and the graphs are safely transferred accordingly. I have several concerns as follows:

(1) I suggest using "LiDAR" instead of "LIDAR" or “lidar” as it is commonly used.

(2) In line 19, the “YawIDs” should be instead by “Yae IDs”.

(3) In line 27, the “waypoints” appeared twice in the keywords.

(4) In lines 57-59, the author should indicate which environments are challenging.

(6) In section “Previous work”, the process to generate topological maps for autonomous vehicles has been significantly addressed, the related works about the waypoint transfer strategy should be introduced in this section.

(7) In section “Conclusions”, the results of experimental analysis results should be quantified.

Overall, my advice is “minor revision”.

Round 2

Reviewer 1 Report

Comments and Suggestions for Authors

Based on the response of the authors, I recognize the value of this paper in the problem of transferring waypoints. It is hoped that the author can explain more about the difference between waypoint transferring and common map matching, and highlight the unique application of waypoint transferring.

Comments on the Quality of English Language

The paper is easy to understand.